# FETT: Expanding the Long Context Understanding Capability of LLMs at Test-Time

## Abstract

Transformer-based Language Models' computation and memory overhead increase quadratically as a function of sequence length. The quadratic cost poses challenges when employing LLMs for processing long sequences. In this work, we introduce ETT (Extend at Test-Time), method for extending the context length of short context Transformer-based LLMs, with constant memory requirement and linear computation overhead. ETT enable the extension of the context length at test-time by efficient fine-tuning the model's parameters on the input context, chunked into overlapping small subsequences.

We evaluate ETT on LongBench by extending the context length of GPT-Large and Phi-2 up to 32 times, increasing from 1k to 32k tokens. This results in up to a 30% improvement in the model's accuracy. We also study how context can be stored in LLM's weights effectively and efficiently. Through a detailed ablation study, we examine which Transformer modules are most beneficial to fine-tune at test-time. Interestingly, we find that fine-tuning the second layer of the FFNs is more effective than full fine-tuning, leading to a further improvement in the models' accuracy.

## 1 Introduction

Transformers have achieved state-of-the-art performance across a wide range of tasks Vaswani et al. (2023). Nonetheless, their scalability to long sequences is fundamentally constrained by the computational and memory demands of the attention mechanism. In particular, standard attention incurs quadratic complexity $\mathcal{O}(N^2)$ in both computation and memory with respect to sequence length $N$, while even optimized variants such as flash attention exhibit linear memory growth Dao et al. (2022). Moreover, during inference, the key–value (KV) cache expands proportionally with the sequence length, introducing additional memory bottlenecks that further impede efficient long-context processing.

In this work, we investigate Test-Time Training (TTT) Krause et al. (2017) to extend the model's context length at test (inference) time with constant memory requirements and linear computational complexity. TTT updates the model parameters using a loss derived by unlabeled test data, and resets the model parameters to their original value after completing the inference for each test data. We introduce ETT (Extend at Test-Time), which extend the context length at test-time by fine-tuning the model's parameters on the input context, chunked into overlapping subsequences.

*From a memory perspective*, ETT leverages the model's parameters and their ability to memorize the data as persistent memory during inference, resetting them at the end of the process. ETT reduces the computational overhead of transformer based LLMs from quadratic to linear and maintains a constant memory footprint regardless of input length since the model input is limited to fixed chunk size.

Our primarily empirical experiment investigates extending the short-context window of small language models (Phi-2 Javaheripi et al. (2023) and GPT-Large Radford et al. (2019)) by up to 32× at test-time through full fine-tuning. This approach result in a noticeable improvement in LongBench Bai et al. (2024) scores.

While ETT has a constant memory requirement, (full) test-time training incurs a 3× model-size overhead, primarily due to the need to store optimizer states and gradients. This raises an important

question: **Can we efficiently and effectively "memorize" the input context at test-time?** To explore this, we conduct empirical studies focused on two key aspects: (1) which model modules, such as self-attention or feed-forward networks, are most effective to fine-tune, and (2) whether fine-tuning shallower versus deeper layers leads to better performance on long-context understanding tasks.

We conduct an empirical ablation study on fine-tuning FFNs (also known as key-value memories Geva et al. (2021)), keys (the 1st layer in the FFNs), values (the 2nd layer in the FFNs), and attention layers and compare them with full fine-tuning. We compare those methods in various long-context understanding tasks and generally observe the superiority of fine-tuning keys over other modules, including full fine-tuning. In fact, we observe that TTT on only key parameters improves the model accuracy while substantially reduces the learnable parameters.

We also empirically evaluate the effectiveness of shallower key layers in ETT performance and observe that shallow layers contribute minimally to the overall performance. Our main result is that we can remove a fraction of the shallower layers from Test-Time Training parameters with minimal degradation in downstream Long Context Understanding benchmarks. This finding allows us to reduce the overhead of applying TTT by freezing the shallow layers and avoiding back-propagation through a portion of layers.

To summarize, our contributions are the following:

- We propose ETT , an architecture-agnostic method that extends the context length of short context pretrained language models at test-time with constant memory and linear computation overhead.

- Through ablation studies, we find that fine-tuning only the first layer of FFN modules (key layer) is more effective than full model tuning, reducing the overhead while improving the performance. Furthermore, we show that training only the top layers of the model preserves performance while reducing compute and memory costs.

The rest of this paper is organized as follows: Section 2 provides some context about the related work. Section 3 describes ETT in detail. In Section 4, we highlight the experiments, and finally, we conclude our findings in Section 5.

## 2 RELATED WORK

Several efforts have been made to overcome the quadratic memory bottleneck in Transformers. Sparse attention mechanisms selectively limit which tokens should participate in self-attention, reducing the complexity from quadratic to linear or sub-quadratic levels depending on the sparsity pattern Child et al. (2019); Beltagy et al. (2020). While sparse attention-based methods can successfully increase the context length by reducing the complexity, they rely on predefined attention patterns. Kernel-based methods Katharopoulos et al. (2020) address the challenge of quadratic complexity by approximating the Softmax function in self-attention with a kernel function, enabling attention computation with linear complexity. However, despite their efficiency, kernel-based methods fall short of Softmax attention both in terms of accuracy and training stability Qin et al. (2022). Alternative architectures to Transformers, including recurrent architectures such as State Space Models (SSMs) Gu & Dao (2024) and State Space Duality (SSD) Dao & Gu (2024), have been proposed to address the quadratic costs at the architectural level and enable scalable evaluation over long-contexts with linear complexity. However, these models often suffer from limited expressiveness Chen et al. (2025) due to their fixed-size hidden states, which constrains their ability to capture complex dependencies and ultimately leads to lower accuracy compared to Transformers in long-context evaluation.

TTT has a long-standing history in the field of machine learning Hinton (1987); Bottou & Vapnik (1992); Schmidhuber (1992). Recently, TTT has been revisited by researchers to be applied to language modeling Ba et al. (2016); Hübotter et al. (2025); Sun et al. (2025); Hardt & Sun (2024); Mahdi Moradi et al. (2025). The basic approach is to directly fine-tune a language model on the test sequence to learn the local probability distribution. Dynamic Evaluation Krause et al. (2018) fine-tunes the model parameters during training with a next-word prediction objective function and substantially improves the model's perplexity. However, it requires over three times the computational cost compared to standard inference. Authors in Clark et al. (2022) improve the efficiency of

Dynamic Evaluation by adding a linear layer, called Fast Weight Layer (FWL), on top of the existing transformer models and only fine-tuning the FWL at test-time. While Dynamic Evaluation and FWL has shown perplexity improvements, their performance on downstream tasks remains unexplored. In this work, we explore the effectiveness of TTT for improving the long-context understanding capabilities of large language models (LLMs) with constant memory requirement.

In a concurrent work, LIFT Mao et al. (2025) proposed memorizing the context in a specialized Gated Memory and utilizing auxiliary tasks, handcrafted for each downstream task, to fine-tune the model at test-time and improve LLMs' long-context performance. In contrast, ETT fine-tunes a subset of the model parameters using a next-word prediction objective function and empirically demonstrates that TTT can effectively and efficiently improve the LLM's long-context understanding capability without the need for external memory or auxiliary task design.

## 3 METHOD

At test (inference) time, given a prompt consisting of an instruction $I$ and a long context $X$, ETT fine-tunes the pretrained model with parameter $\theta_0$ on the long context $X$ and implicitly memorizes the sequence in the model parameters. To address the quadratic computation overhead and memory footprint of transformer based models, ETT chunks long context $X = (t_0, t_1, \ldots, t_L)$ into subsequences $\{s_0 = t_{0:n}, s_1 = t_{n:2n}, \ldots\}$, with fixed length of $n$ tokens. The subsequences are randomly grouped into batches, with batch $i$ (zero-indexed) denoted as $b_i$, and fine-tuned using a next-word prediction objective function to edit the model's implicit knowledge.

The pretrained model parameters are used to compute the log probability of the first batch $\sum_{s_i \in b_0} \log p(s_i|\theta_0)$. This probability is then employed to calculate the cross-entropy loss $L(b_0)$ and the corresponding gradient $\nabla L(b_0)$. The gradient $\nabla L(b_0)$ is subsequently used to update the model, resulting in the adapted parameters $\theta_1$. This process is repeated for the second batch, where the probability $p(b_1|\theta_1)$ is evaluated, and the procedure is carried out iteratively for the remaining batches (See Algorithm 1).

---

**Algorithm 1** ETT Algorithm

---

1: **Input:** Pretrained model $\mathcal{M}$ with parameters $\theta_0$, Context $X$, Instruction $I$, number of TTT epochs E.
2: Decompose $X$ into subsequences: $\{s_0 = t_{0:n}, s_1 = t_{n:2n}, \ldots\}$
3: **for** each epoch e $\in [1 \ldots E]$ **do**
4:     Randomly group subsequences into batches, batch $i$ denoted as $b_i$
5:     **for** each batch $b_i$ **do**
6:         $\mathcal{M}_{\theta_e}$ = fine-tune model $\mathcal{M}_{\theta_{e-1}}$ using a next-word prediction objective function on the current batch
7:     **end for**
8: **end for**
9: Sample answer $A$ from $p_{\theta_E}(.|I)$
10: Reset the parameters to their original values in $\theta_0$
11: **return** $A$

---

## 4 EXPERIMENTS

We evaluate ETT on GPT-Large and Phi-2. To thoroughly evaluate its ability to handle long-context sequences, we use LongBench Bai et al. (2024), which comprises 21 real-world and synthetic long-context tasks.

We begin by examining the improvements in long-context capabilities of the studied models with ETT and full fine-tuning at test-time. Next, we investigate whether the test-time training overhead can be reduced. Specifically, we demonstrate that: 1) Fine-tuning only the up-projection layers in the feed-forward networks (also known as key Geva et al. (2021)) can further improve accuracy compare to full fine-tuning while reducing the number of trainable parameters by approximately 70%. 2) We find that restricting fine-tuning to only the deeper layers allows us to reduce the number

of trainable parameters at test-time to just 15% of the model's parameters, with little to no loss in performance.

**Experimental details.** In all of the experiments, we chunk the long-context input into subsequences of 512 tokens with an overlap of 32 tokens between the adjacent chunks. For each input, we fine-tune the model for 10 epochs and restore the original model parameters after running inference. We adopt the Adam optimizer with a learning rate of $5e^{-4}$ and weight decay of $0.5$.

## 4.1 ETT ENHANCES LONG-CONTEXT UNDERSTANDING ACROSS STANDARD LONG-CONTEXT TASKS

Figure 1 shows the impact of ETT on the long-context understanding capabilities of Phi-2 and GPT-Large plotted as a function of the context length. In all of the experiments, the context $X$ is truncated in the middle following Bai et al. (2024). We applied full fine-tuning at test-time and reported the average LongBench score across all 21 tasks. We observe that the performance consistently improves across all LongBench tasks as the context length increases. Our experiments were conducted on a 8 NVIDIA V100 GPUs with 32GB HBM2 memory, as the memory footprint remains constant across different context window sizes. We estimate the training FLOPs for Full Fine Tuning following Kaplan et al. (2020).

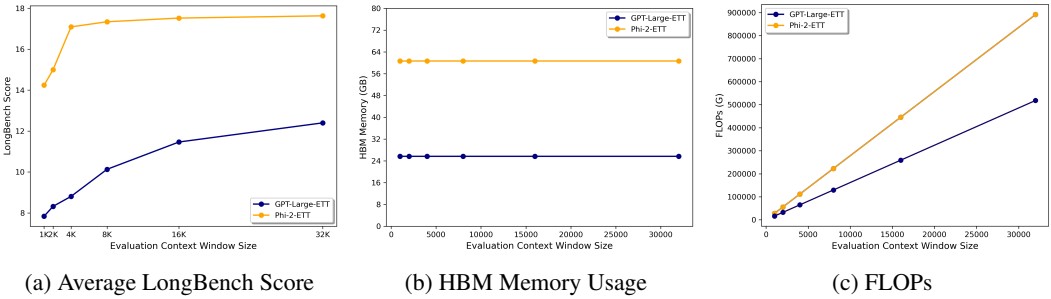

(a) Average LongBench Score  (b) HBM Memory Usage  (c) FLOPs

Figure 1: Average LongBench score (%), HBM memory consumption (GB), and FLOPs (G) under different truncation sizes. ETT extends the context window of Phi-2 and GPT-Large by up to 16× and 32×, respectively. Performance improves with longer context lengths while maintaining constant memory usage and only linear growth in computation.

## 4.2 SELECTIVE FINE-TUNING AT TEST-TIME OUTPERFORMS FULL FINE-TUNING

In this work, we conduct an empirical ablation study to evaluate the effectiveness of selectively fine-tuning different modules in enhancing long-context understanding at test-time. Specifically, we fine-tune individual modules of the model: the keys (i.e., the first linear layer in the FFN, denoted as $\text{FFN}_{\text{Up}}$), the values (i.e., the second linear layer in the FFN, denoted as $\text{FFN}_{\text{Up}}$), and the attention parameters (i.e., the key, query, and value projections: K, Q, V). We compare these strategies based on their impact on ETT 's performance.

*This experiment aims to provide insights into the effectiveness of fine-tuning different modules at test-time.*

As shown in Table 1, fine-tuning $\text{FFN}_{\text{Up}}$ consistently outperforms other strategies across various settings. In particular, fine-tuning $\text{FFN}_{\text{Up}}$ instead of applying full fine-tuning improves the Long-Bench score from 11.30 to 12.57 for GPT-Large, and from 16.75 to 18.3 for Phi-2 while reducing the number of trainable parameters—and consequently the memory footprint—by 70%. This observation aligns with previous studies, which have shown that updating the keys within FFNs leads to performance improvements compared to updating the values when tuning LLMs for knowledge editing task Qiu et al. (2024).

| ETT Target | GPT-Large Radford et al. (2019) | | Phi-2 Javaheripi et al. (2023) | |
|---|---|---|---|---|
| | **Trainable** | **LongBench Score** | **Trainable** | **LongBench Score** |
| Full Fine-Tuning | 100.0 % | 11.30 | 100.0 % | 17.33 |
| FFN | 60.99 % | 11.81 | 60.37 % | 17.21 |
| $FFN_{Up}$ | 30.48 % | **12.57** | 30.19 % | **18.33** |
| $FFN_{Down}$ | 30.50 % | 11.15 | 30.18 % | 16.75 |
| $Attention_{QKV}$ | 30.48 % | 11.11 | 30.19 % | 18.31 |
| Baseline | 0 % | 9.58 | 0 % | 15.04 |

Table 1: ETT Target and corresponding LongBench scores for Experiment GPT-Large and Phi-2.

### 4.3 SHALLOWER KEY LAYERS ARE LESS EFFECTIVE THAN THE DEEPER ONES

We also empirically investigate the effectiveness of fine-tuning shallower $FFN_{Up}$ layers at test-time. If we freeze a block of shallow layers and observe no impact on ETT 's performance, it suggests that those layers are not essential for ETT . To identify the optimal block of shallow layers to freeze, we incrementally freeze blocks of shallow layers and evaluate ETT 's performance at each step. This bottom-up strategy reduces the number of trainable parameters and computational cost as backpropagation is not required for the contiguous block of shallow, frozen layers.

Figure 2 shows ETT 's average LongBench score as the fraction of shallow key ($FFN_{Up}$) layers frozen. We observe that fine-tuning only the top 80% of $FFN_{Up}$ layers achieves similar performance as fine-tuning all layers. Importantly, there is a sharp performance degradation when freezing more than 40% of the shallow layers, indicating a transition point beyond which key contextual information is no longer preserved.

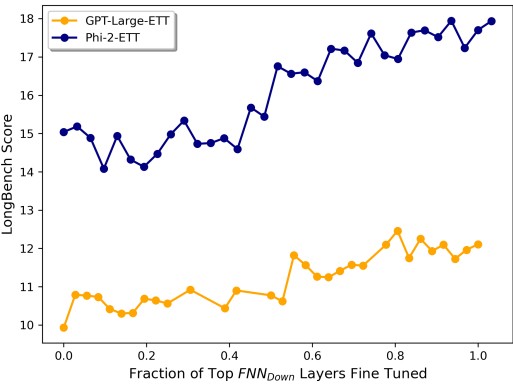

Figure 2: ETT 's LongBench score as a function of the fraction of deep $FFN_{Up}$ layers fine-tuned. We can store the long input in the parameters of the top 80% of $FFN_{Up}$ layers without significant performance degradation.

The LongBench scores for GPT-Large and Phi-2, with and without the parameter-efficient version of ETT , are reported in Tables 2 and 3. In all the experiments, we fine-tuned the top 80% of the $FFN_{Up}$ layers.

| | Single-Doc QA | | | | Multi-Doc QA | | | | Summarization | | | |
|---|---|---|---|---|---|---|---|---|---|---|---|---|
| | MultiFieldQA-zh | MultiFieldQA-en | NarrativeQA | Qasper | 2WikiMultihopQA | HotpotQA | MuSiQue | DuReader | MultiNews | GovReport | QMSum | VCSUM |
| GPT-Large | **6.7** | **13.36** | 2.2 | 5.29 | **9.3** | 5.55 | 3.1 | 14.19 | 21.62 | 19.01 | 13.41 | 8.67 |
| GPT-Large-ETT | 5.62 | 12.07 | **2.7** | **7.22** | 8.52 | **6.47** | **4.54** | **17.36** | **24.81** | **25.99** | **13.96** | **9.53** |
| Phi-2 | **19.42** | 34.9 | **13.12** | 6.94 | 9.27 | 12.3 | 9.14 | 6.54 | **32.52** | 26.32 | 18.09 | **11.06** |
| Phi-2-ETT | 18.37 | **37.76** | 9.46 | **9.64** | **13.3** | **18.26** | **9.26** | **20.2** | 25.5 | **34.01** | **19.3** | 10.55 |

Table 2: LongBench score comparison between GPT-Large and Phi-2, with and without ETT (selectively fine-tuned) on Single-Doc QA, Multi-Doc QA, and Summarization tasks.

| | Few-shot Learning | | | | Synthetic Tasks | | | Code Completion | | |
|---|---|---|---|---|---|---|---|---|---|---|
| | TriviaQA | SAMSum | TREC | LSHT | PassageRetrieval-en | PassageRetrieval-zh | PassageCount | LCC | RepoBench-P | Avg |
| GPT-Large | **10.65** | 22.12 | 22.83 | 0 | **3.47** | **2.5** | 0.87 | 9.12 | 13.06 | 9.85 |
| GPT-Large-ETT | 7.56 | **24.42** | **27.98** | **4.5** | 3.33 | 1.83 | **1.9** | **25** | **22.52** | **12.27** |
| Phi-2 | **2.38** | 3.03 | 28.57 | **23.15** | **14.29** | 4.76 | **1.59** | **18.05** | 20.52 | 15.04 |
| Phi-2-ETT | **2.38** | 15.38 | 42.86 | 22.53 | **14.29** | 19.05 | 0 | 16.1 | **22.29** | 18.34 |

Table 3: LongBench score comparison between GPT-Large and Phi-2, with and without ETT (selectively fine-tuned) on Few-shot Learning, Synthetic, and Code Completion tasks.

## 4.4 ETT ENABLES PHI-2 TO COMPETE WITH 8B LLMs FINE-TUNED ON LONG CONTEXTS, USING CONSTANT MEMORY AND LINEAR COMPUTATION

We further compare ETT against several popular baselines on the long-context benchmarks, including fixed-length models fine-tuned on long-context data (Vicuna1.5-7B-16k[1], LongChat1.5-7B-32k[2], together/llama-2-7b-32k[3], Llama-3-8B-Instruct-Gradient-1M[4]), as well as context-extension methods such as SelfExtend Jin et al. (2024) and LIFT Mao et al. (2025). We benchmark the models on five long-context tasks from LongBench, using the same tasks reported in LIFT.

Across almost all tasks, when applied to Phi-2, ETT consistently outperforms context-extension methods. Specifically, on PassageRetrievalEN and Musique, Phi-2-ETT achieves 14.29 and 9.26, respectively, substantially exceeding the performance of Phi-2-LIFT (8.17, 3.96) and Phi-2-SelfExtend (2.38, 3.89).

ETT also allows Phi-2 to achieve competitive results relative to 8B-parameter models despite having significantly fewer parameters. Notably, ETT improves Phi-2's performance on GovReport from 26.32 to 34.01, surpassing all studied baselines.

Overall, these results confirm that ETT substantially enhances the long-context capability of Phi-2, outperforming SelfExtend and LIFT on several key benchmarks, and achieving performance competitive with much larger LLMs.

---

[1] https://huggingface.co/lmsys/vicuna-7b-v1.5-16k
[2] https://huggingface.co/lmsys/longchat-7b-v1.5-32k
[3] https://huggingface.co/togethercomputer/LLaMA-2-7B-32K
[4] https://huggingface.co/gradientai/Llama-3-8B-Instruct-Gradient-1048k

| Methods | Musique | Narrativeqa | Qmsum | GovReport | PassageRetrievalEN |
|---|---|---|---|---|---|
| Phi-2-ETT  (ours*) | **9.26** | 9.46 | **19.30** | **34.01** | **14.29** |
| Phi-2-LIFT | 3.96 | 11.78 | 15.32 | 29.39 | 8.17 |
| Phi-2-Se | 3.89 | **12.04** | 14.58 | 27.90 | 2.38 |
| LLaMa3-8B-32k-LIFT | 10.99 | **25.84** | **22.96** | **31.26** | **41.67** |
| LLaMa3-8B-32k-Se | 3.89 | 12.04 | 14.58 | 27.90 | 2.83 |
| Llama-3-8B-Instruct-Gradient-1M | **13.89** | 12.04 | 14.58 | 27.90 | 2.83 |
| together/llama-2-7b-32k | 6.19 | 15.65 | 17.18 | 29.28 | 23.0 |
| Vicuna1.5-7B-16k | 9.8 | 19.4 | 22.8 | 27.9 | 4.5 |
| LongChat1.5-7B-32k | 9.7 | 16.6 | 22.7 | 30.08 | 30.50 |

Table 4: Performance comparison of different LLMs on LongBench. The number (e.g., '25k') denotes the maximum input length. The postfixes Se, LIFT, and ETT indicate that SelfExtend Jin et al. (2024), LIFT Mao et al. (2025), and ETT (our method), respectively, are applied to the corresponding model. LongChat1.5-7B-32k , together/llama-2-7b-32k and Vicuna1.5-7B-16k are fixed-length models fine-tuned on long contexts Jin et al. (2024). The best performance is highlighted in bold. ETT enhances the long-context understanding of Phi-2 (2.7B parameters) to a level competitive with models up to 8B parameters, while maintaining constant memory requirements and linear computation with respect to context length. Notably, ETT improves Phi-2's performance on GovReport from 26.32 to 34.01, surpassing all studied baselines.

## 5 CONCLUSION AND FUTURE WORK

In this work, we introduce ETT , an architecture-agnostic, lightweight and efficient approach for extending the context length of pretrained language models at inference time with constant memory and linear computation overhead. Our method enables transformer based language models, such as GPT-Large and Phi-2, originally trained with short context windows to process significantly longer inputs. ETT demonstrates consistent improvements in long-context understanding across multiple tasks from LongBench. We also investigated the effectiveness of different transformer modules and shallow-layer in test-time training. Specifically, we demonstrated that: 1) Fine-tuning only the up-projection layers in the feed-forward networks improves ETT accuracy compared to full fine-tuning while reducing the number of trainable parameters by approximately 70%. 2) We showed that restricting fine-tuning to only the deeper layers allows us to reduce the number of trainable parameters at test-time to just 15% of the model's parameters, with little to no loss in performance. Our results highlight the effectiveness of ETT , offering a practical solution for scaling LLMs to longer sequences.

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
