# OpenReview forum: "ETT: Expanding the Long Context Understanding Capability of LLMs at Test-Time"
_ICLR.cc/2026/Conference — ICLR 2026 Conference Withdrawn Submission_

### Official Review · Reviewer_7YA3 · 2025-10-28

**Soundness:** 2
**Presentation:** 3
**Contribution:** 2
**Rating:** 4
**Confidence:** 4

**Summary:**

This paper introduces ETT, a method to expand the long-context understanding capability of transformer-based language models at inference by chunking the input into overlapping subsequences and fine-tuning model parameters on the fly via test-time training. The approach claims constant memory and linear computational overhead, enabling small models (e.g., GPT-Large, Phi-2) to process sequences up to 32x their original length. A thorough set of experiments is presented on LongBench, including ablations over which layers and modules should be fine-tuned, and comparative results against context-extension and fine-tuned long-context LLMs.

**Strengths:**

ETT addresses a pressing practical bottleneck in using LLMs for long-context tasks, trading off memory inefficiency for test-time adaptation without architectural modification.

**Weaknesses:**

1. The overall contribution of the paper shows limited originality. The proposed ETT method merely applies the existing Test-Time Training concept to long-context inference without introducing substantial theoretical analysis or methodological innovation.
2. The review of classic long-context Transformer variants is missing, such as Compressive Transformer [1], BigBird [2], and Transformer-XL [3].
3. The paper does not compare its models with recent state-of-the-art long-context large models such as Qwen2.5 [4] and GLM-4 [5], which makes it difficult to convincingly demonstrate the competitiveness of the proposed method.
4. Experiments are conducted only on Phi-2 (2.7B) and GPT-Large (774M) which are relatively outdated and small compared to contemporary large-scale LLMs. It is recommended to applies ETT on modern large models to validate its effectiveness rather than relying solely on small baselines.
5. The paper title uses the term “FETT,” while the proposed method is consistently referred to as “ETT” throughout the main text.
6. Figure 1 is insufficiently described.
7. Table 4 is not explicitly referenced or analyzed in the paper.

[1] Rae J W, Potapenko A, Jayakumar S M, et al. Compressive transformers for long-range sequence modelling[J]. arXiv preprint arXiv:1911.05507, 2019.
[2] Zaheer M, Guruganesh G, Dubey K A, et al. Big bird: Transformers for longer sequences[J]. Advances in neural information processing systems, 2020, 33: 17283-17297.
[3] Dai Z, Yang Z, Yang Y, et al. Transformer-xl: Attentive language models beyond a fixed-length context[J]. arXiv preprint arXiv:1901.02860, 2019.
[4] Team Q. Qwen2 technical report[J]. arXiv preprint arXiv:2407.10671, 2024, 2(3).
[5] GLM T, Zeng A, Xu B, et al. Chatglm: A family of large language models from glm-130b to glm-4 all tools[J]. arXiv preprint arXiv:2406.12793, 2024.

**Questions:**

1. Can ETT achieve similar or better performance on larger models, such as 7B or 32B？
2. Does the fine-tuning during inference introduced by ETT significantly increase the model's inference latency, especially for longer sequences?

---

### Official Review · Reviewer_uQ55 · 2025-10-28

**Soundness:** 3
**Presentation:** 3
**Contribution:** 2
**Rating:** 2
**Confidence:** 4

**Summary:**

This work introduces a test-time compute algorithm that expands the effective context length of LLMs by chunking the input sequence as batches, which are then used to finetune the model weights. PETT expands GPT-Large and Phi-2 from 1k to 32k context, tested on LongBench suite. On top of that it also studies freezing and finetuning different modules across different layers and found certain layers to be more effective when finetuned with a single test sample.

**Strengths:**

1. The paper compare several different baselines including LIFT, SelfExtend
2. It chooses a decently long generative tasks to evaluate (i.e. LongBench)
3. It proposes a novel idea of finetuning a single test sample before generation, which leads to accuracy improvement.

**Weaknesses:**

1. The baseline is rather weak as a “long context” model.
2. The experiments use 10 epochs to finetune the model for every single input, whose cost is not well studied and explained. The reviewer personally thinks this as less viable as a method for large models.
3. Both GPT-large and Phi-2 have not used any out-of-the-box tricks for context extension (e.g. tuning the RoPE parameters as suggested by https://arxiv.org/abs/2309.16039) and should be at least compared with changing the RoPE and finetune on all (or a subset of) input sequences (see the same amount of data)
4. Although the study on which modules to finetune results provides insights, it is not well tested against larger or other model families. The heuristics clearly need more experiments to support the finding. Otherwise, it’s really unfair to try all different cases and choose the best one, which itself requires not only computation but also seeing the entire test inputs.
5. Table 4 should include all LongBench tasks for completeness if possible.
6. Most LongBench tasks are well short of 32k, and should be specified as the claim is to expand from 1k to 32k.
7. There are many other methods that can compress the prompt before sending to the model, which implicitly expands the effective context length and should be at least cited properly (e.g. LLMLingua https://arxiv.org/abs/2310.05736 , SpecPrefill https://arxiv.org/abs/2502.02789 , and GemFilter https://arxiv.org/abs/2409.17422)

**Questions:**

Listed above in the weakness section.

---

### Official Review · Reviewer_xQiZ · 2025-10-29

**Soundness:** 2
**Presentation:** 1
**Contribution:** 1
**Rating:** 2
**Confidence:** 4

**Summary:**

The paper proposes a method to enhance the long-context understanding of transformer-based language models without retraining or increasing memory usage. The key idea is to perform test-time fine-tuning on the input sequence, chunked into overlapping subsequences, so that the model can “memorize” the long context directly in its parameters during inference.

The method is lightweight, architecture-agnostic, and exhibits constant memory consumption and linear computational complexity with respect to input length.

Experiments on LongBench show that extending GPT-Large and Phi-2 from 1k to 32k context windows improves accuracy by up to 30%, and that fine-tuning only the second FFN layer (key/value module) yields better results than full fine-tuning while reducing trainable parameters by ~70%.

**Strengths:**

1. ETT offers a plug-and-play mechanism for improving long-context performance on existing models (e.g., Phi-2, GPT-Large) without re-training or additional memory, making it potentially valuable for deployment in resource-constrained environments.

**Weaknesses:**

1. Experiments are restricted to small models (Phi-2, GPT-Large) and a single benchmark (LongBench). Results on stronger baselines (Llama3, Mistral, Qwen2.5) and more diverse tasks would significantly strengthen the paper.   Small models often fail to show the same qualitative behavior in memory usage, optimization stability, or attention saturation seen in larger LLMs. More benchmark is also needed.

2. This paper lacks baselines. Other methods like LongLoRA, Position Interpolation are typically validated on Vicuna-7B, LLaMA-2/3-7B, or Mistral-7B and should also be compared.

2. The paper is short (~7 pages), lacks detailed methodology (e.g., FLOPs measurement, convergence analysis), and has no appendix or reproducibility statement. Overall, it reads more like a technical report or workshop submission than a full ICLR paper.

3. The motivation of this method is unclear.

**Questions:**

See weakness.

---

### Official Review · Reviewer_Exp4 · 2025-11-01

**Soundness:** 2
**Presentation:** 1
**Contribution:** 1
**Rating:** 2
**Confidence:** 4

**Summary:**

This paper proposes ETT (Extend at Test-Time), a method for extending the context length of short-context transformer-based LLMs at test time. The approach chunks long input sequences into overlapping subsequences of fixed length, then fine-tunes model parameters on these chunks using next-word prediction. The authors evaluate on GPT-Large and Phi-2, extending context from 1k to 32k tokens on LongBench tasks. Through ablation studies, they find that fine-tuning only the FFN up-projection layers outperforms full fine-tuning while using 70% fewer parameters, and that fine-tuning only the top 80% of layers maintains performance while reducing computation.

**Strengths:**

1. The method achieves constant memory footprint and linear computational complexity with respect to context length.
2. The systematic investigation of which modules to fine-tune (FFNs, attention, specific layers) provides valuable insights.
3. The paper provides sufficient implementation details (chunk size, overlap, learning rate, etc.) for reproducibility.

**Weaknesses:**

1. The paper has limited novelty. Test-time training for language models is well-established. The primary contribution is applying this to long-context understanding, which is incremental.
2. Fine-tuning for 10 epochs at test time for every input incurs massive computational overhead, making the method not practical in real-world use cases.
3. The testing scope is very limited. Only two small models are evaluated, without any modern long-context technique baselines like the PE extrapolation methods.

**Questions:**

1. The connection between "keys" in FFNs and the key-value memory interpretation/key in self-attention is very confusing and needs a clearer explanation.
2. Please use \citep instead of \citet where necessary.
3. Why chunk the sequence during finetuning? If you chunk the sequence, how do you adapt the model to longer sequences that require long-range dependency?
4. How do you assign position IDs for your instance during finetuning?
5. It would be beneficial if some more modern models (like the Llama 3 series or Qwen-2.5) could be evaluated.
6. What is the "full-finetuning" setting?
7. The improvements shown in Tables 2 and 3 are not consistent. Is there a reason?

---

### Note · Authors · 2025-11-12

**Comment:**

I would like to thank the reviewers for their insightful feedback and withdraw the paper submission.

**Withdrawal Confirmation:**

I have read and agree with the venue's withdrawal policy on behalf of myself and my co-authors.